# Application of Machine Learning Techniques for Predicting Potential Vehicle-to-Pedestrian Collisions in Virtual Reality Scenarios

Ángel Losada [1],*, Francisco Javier Páez [1], Francisco Luque [2] and Luca Piovano [2]

[1] Department of Accidentology, University Institute for Automobile Research Francisco Aparicio Izquierdo (INSIA-UPM), Universidad Politécnica de Madrid, 28031 Madrid, Spain

[2] Center for Energy Efficiency, Virtual Reality, Optical Engineering and Biometry (CEDINT-UPM), Universidad Politécnica de Madrid, 28223 Pozuelo de Alarcón, Spain

* Correspondence: angel.losada.arias@upm.es; Tel.: +34-616-909-632

**Abstract:** The definition of pedestrian behavior when crossing the street and facing potential collision situations is crucial for the design of new Autonomous Emergency Braking systems (AEB) in commercial vehicles. To this end, this article proposes the generation of classification models through the deployment of machine learning techniques that can predict whether there will be a collision depending on the type of reaction, the lane where it occurs, the visual acuity the level of attention, and consider the most relevant factors that determine the cognitive and movement characteristics of pedestrians. Thereby, the inclusion of this type of model in the decision-making algorithm of the AEB system allows for modulating its response. For this purpose, relevant information on pedestrian behavior is obtained through experiments made in an ad-hoc, Virtual Reality (VR) environment, using a portable backpack system in three urban scenarios with different characteristics. Database generation, feature selection, and k-fold cross-validation generate the inputs to the supervised learning models. A subsequent analysis of the accuracy, optimization, error measurement, variable importance, and classification capability is conducted. The tree-based models provide more balanced results for the performance metrics (with higher accuracy for the single decision tree case) and are more easily interpretable and adaptable to the algorithm. From them it is deduced the high importance of the reaction type and the relative position where it occurs, coinciding with the high significance of these factors in the analyzed collisions.

**Keywords:** pedestrian behavior; machine learning techniques; virtual reality; accident risk

## 1. Introduction

In the 2010–2020 period, pedestrians represented 21% of the total number of people killed in Spanish road traffic accidents [1]. Considering the figures for fatalities in urban road accidents, the number of pedestrians amounted to 50% of the deaths recorded in cities, which is indicative of the significance of this group in urban environments as unprotected road users.

Therefore, pedestrians are a group of interest not only when it comes to implementing measures to increase road safety in cities, but also to the development of Advanced Driver Assistance systems (ADAS) capable of recognizing and predicting critical situations and taking active countermeasures to avoid or significantly reduce the accident rate in this group.

To reproduce realistic conditions involving potential pedestrian-vehicle impacts, these scenarios can be simulated using Virtual Reality (VR) techniques [2], which makes it possible to replicate potential run-over events more safely and flexibly.

The generation of a predictive model that can evaluate if a collision is likely to occur in a possible hit-and-run situation has several practical applications. Such models may

help the optimization of the decision algorithm of AEB systems and Autonomous Vehicles (AV). In this context, the authors of [3] identified the main areas for improvement of an AEB system onboard a commercial vehicle through the assessment of the most recurrent avoidance reactions in VR tests and the corresponding dynamic simulation. Therefore, the focus is on determining the optimal set of explanatory variables that, in accordance with the information collected by the sensors of such onboard systems, may improve the estimation of the likelihood of a collision.

## 2. State of the Art

To increase the validity and generalizability of the results obtained from user interaction with the virtual environment, the 3D scenario should be as realistic and immersive as possible. Indeed, predominant technologies in the use of virtual reality techniques, such as Head-Mounted Displays (HMDs), offer greater portability and stereoscopic rendering capabilities, leading to superior user immersion with the interface [4,5].

In conjunction with other VR-related technologies, it is possible to easily collect physiological and/or biometric data. The deployment of motion capture [4] allows for recording the movements of the user's limbs and other body parts for their subsequent inclusion and analysis in the digital animation of an avatar. The use of portable backpack devices extends the physical limits of the areas where users are allowed to move, resulting in more natural behavior. Other external tracking systems may complement the collation of relevant data to different degrees of detail (e.g., eye trackers and haptic devices for hand or finger tracking).

The major issue reported by users while using HMD is cybersickness, a general discomfort caused by the discrepancy between simulated visual motion and the perception of movement from the vestibular system [5]. Detecting and quantifying its effects on individuals is important to establish the validity of the user's VR experience. To this end, frameworks such as the Simulator Sickness Questionnaire (SSQ) [6] and the Fast Motion Sickness Scale (FMS) have been proposed. On the other hand, the System Usability Scale (SUS) is used to measure usability in VR environments. Usability is defined as the effectiveness, intuitiveness, and satisfaction with which specified users can achieve specified goals in particular environments, particularly interactive systems [7].

Pedestrian behavior, in addition to being defined by variables related to their relative movement with respect to the vehicle (position, speed, acceleration, and trajectory), is also strongly associated with visual perception [8–10]. Eye-tracking technologies embedded in Virtual Reality equipment allow the real-time recognition and detection of the objects at which the pedestrian is looking. This information may be employed to identify the main sources of distraction [11] that can compromise the response to a hazard on the road.

Furthermore, visual perception affects the estimation of speeds and distances relative to other objects/vehicles. For instance, depth perception [12] influences the calculation of the time needed to cross a street when an automobile is approaching a crosswalk, since the pedestrian evaluates the distance to that vehicle and the width of the roadway to adjust the crossing speed. Furthermore, ref. [13] demonstrated that pedestrians tend to overestimate Time-to-Collision (TTC, time required for two objects to collide if they continue at their current speed and along the same path) at higher approaching vehicle speeds, resulting in riskier time gaps. This effect is more noticeable in VR scenarios due to the restriction of the field of view (FOV) [14].

Concerning the characterization of the pedestrian's behavior, ref. [15] explores the main crossing patterns at signalized intersections, focusing on the most significant factors affecting pedestrian compliance behavior and the reasons for pedestrian non-compliance, as well as the possibility of interacting with vehicles under certain traffic conditions. Other studies focus on the estimation of reaction times and the observation of different speeds when crossing the road as a function of age and/or gender [16,17].

To this end, the application of Machine Learning Techniques (MLT) is expected to bring new insights regarding which features have the most promising predictive power

to avoid traffic collisions. In this respect, some research is focused on the development of multilevel predictive models for vehicle-to-pedestrian collision avoidance using clustering and classification techniques [18]. Other authors do not analyze the probability of impact occurrence, but the intentionality of the pedestrian when they see an approaching vehicle from the road curbside, considering different approaches to predict the pedestrian path [19].

Machine learning classification techniques are especially worthwhile for estimating pedestrian intention, allowing the identification of the main factors explaining pedestrian movement and reactions, using bagging techniques such as Random Forest [20] and Support Vector Machine (SVM) [21].

Some other studies revolve around the generation of algorithms that are also focused on the development and optimization of emergency braking systems and autonomous vehicles through the generation of computationally scalable decision-making models that provide robustness to uncertainty in the pedestrian state, such as Markov decision processes [22–24].

The most common technology integrated into vehicles for the detection of pedestrians and other vehicles on the road is the fusion sensor, which combines the performance of a camera housed in the windshield and a LIDAR (light detection and ranging) or radar sensor. Regarding pedestrian identification, some recent approaches are based on the application of a unified deep learning model, where different convolutional layers (CNN- Convolutional Neural Network) are combined for feature extraction using the HOG (Histogram of Oriented gradients) technique, part deformation and occlusion handling, and a final SVM classifier [25]. The use of the CNN for image processing continues to be extended in recent works, as in [26], where the pedestrian detection task is performed using a 3D CNN that allows rejecting true false pedestrians through the vehicle fusion sensor; or in [27], with a multitask R-CNN for complex nighttime scenarios, including distance estimation. LIDAR, on the other hand, is more accurate but less economical than radar, and shows certain advantages over the camera, since it is not affected by light conditions and is computationally faster in detecting the distance and direction of the object [28].

Hence, with the development of these safety-oriented technologies, the general study of the movement of VRU (Vulnerable Road Users, including pedestrians, cyclists, motorcycle riders, users of personal mobility devices, children seven years and under, and elder people) is being considered to enhance the effectiveness and reliability of these systems. Thus, intending to reduce potential crashes, some surveys focus on understanding the limb motions [29], and the corresponding comparison between crossing and non-crossing cases. For the case of fully autonomous vehicles, some models based on perception-decision-action models [30] have demonstrated a certain level of natural harm-avoidance result, by calculating the possibility of occurrence of a vehicle-to-pedestrian collision as a function of human avoidance capability.

*Proposed Work*

This research addresses the study of how pedestrians behave before potential collision situations and the characterization of this behavior through Virtual Reality tests, which allow for simulating accidents in urban environments. Moreover, the analysis of the data obtained through VR simulations and characterizing the pedestrian–vehicle interaction will allow the generation of supervised Machine Learning models to classify and predict whether a collision is likely to occur as a function of the pedestrian's movement and degree of visual attention and depth perception. The application of different classification techniques may lead to distinct results so that their evaluation makes it plausible to compare them not only in terms of the metrics that define the accuracy and predictive capacity of classes but also from the feasibility and the possibility of being implemented in a prototype of a new generation AEB system.

The novelty of this approach is to obtain more realistic and reliable pedestrian crossing behavior data due to the high immersive capability of VR scenarios, which allow the user to interact with an environment with characteristics and conditions similar to reality.

Additionally, the attention factor is introduced, through the rotation of the subject's head. In this way, the classification algorithms work with less biased information and establish a basis for their implementation in AEB devices and their technological advancement, due to the easy interpretation and accessibility of the variables.

The structure of this paper can be broken down into the ensuing sections: Section 3 illustrates the methodological scheme followed (Section 3.1), the generation of the Virtual Reality scenarios and the equipment and facilities used for the tests (Section 3.2), the procedure performed in the experimental session (including the sample selection, test guideline and questionnaire results) (Section 3.3), the database generation (Section 3.4), the data preprocessing and variable analysis (Section 3.5); Section 4 includes the questionnaire results (Section 4.1), the description of the Machine Learning models and techniques used, and the model fitting of each of them (Section 4.2); Section 5 compares the models in terms of accuracy and predictive capacity of each method; and Section 6 summarizes the content of the article and establishes future lines of this research.

## 3. Materials and Methods

This section shows the methodology conducted for the elaboration of this research. It also describes how the previous study of real accidents was carried out to define the scenarios of the experimental VR session, the reconstruction of these scenarios using graphic design tools, and the description of the VR technology (equipment, facility, and type of recordings). On the other hand, the sample of users of the experimental session is also outlined and, with the aim of validating the testing procedure, the content of the questionnaires used to evaluate the degree of physical affectation, usability, and the level of immersion during the tests is detailed. Finally, the generation of the database is explained by means of graphic analysis and calculation of the main variables, data preprocessing, feature selection to obtain the explanatory variables, and graphic evaluation of the level of correlation of these predictors with respect to the output variable of the models.

### 3.1. Methodology

The methods presented in this section integrate three clearly differentiated parts (Figure 1): the generation of hit-and-run scenarios through a detailed study of traffic accidents under specific conditions (green block), the definition of tests, the choice of a sample of subjects for testing in virtual environments (blue block), and the generation of Machine Learning classification models to define pedestrian behavior and its relationship with the possibility of being hit by a car in potential collision situations (orange block).

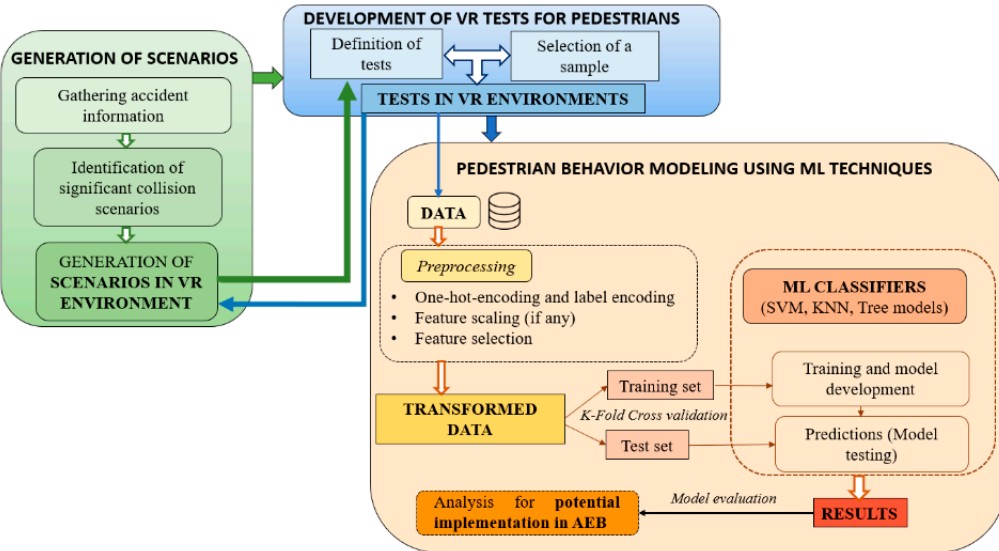

**Figure 1.** Methodology main scheme of the pedestrian behavior modeling procedure.

### 3.2. VR Environment and Equipment

The identification and design of the main urban run-over scenarios were conducted through a comprehensive retrospective in-depth study of accidents involving VRU in the city of Madrid, Spain. From a database provided by INSIA-UPM, a total of 100 accidents in which a vehicle knocks down a pedestrian have been extracted. Urban roads with more than one lane in each direction of traffic have been chosen. This makes it possible to evaluate the reaction of the pedestrian on different lanes by incorporating restrictions on visibility due to the presence of parked vehicles. Taking into account the INSIA-UPM database, in 80% of the cases the vehicle approaches the pedestrian from the left side, so this will be the configuration modeled in the VR scenarios.

In the analyzed database, three types of scenarios were identified: T1, collision at a traffic light regulated crosswalk (28%); T2, collision at a non-traffic light regulated crosswalk (21%), and T3, collision outside a crosswalk (51%). Of these, the first type of scenario is the most significant, since the impact vehicle speed was higher than 40 km/h in most cases, and the risk of death by collision is higher than 50% above this speed [31].

Considering those belonging to class T1 and whose collisions occurred at impact speeds above 40 km/h, the following urban scenarios are identified and modeled in VR (Table 1):

**Table 1.** Scenarios modeled in VR for the experimental session.

| Scenario | Street Lane for VR Collision Point | Side from Which the Vehicle Appears | GPS Coordinates |
|---|---|---|---|
| **Av. de los Toreros (SC1)** | First | Left | 40°25′54.8″ N 3°39′41.8″ W |
| **Av. Machupichu (SC2)** | Second | Left | 40°27′38.6″ N 3°37′57.7″ W |
| **Calle Hermanos García Noblejas (SC3)** | Third | Left | 40°25′47.7″ N 3°37′56.3″ W |

The reconstruction of the aforementioned scenarios has been carried out using Unity3D software (Figure 2). In this powerful cross-platform development framework, the realistic 3D modeling of the traffic sceneries (including both static and dynamic elements of the streets) has been integrated with modules and elements simulating the traffic logic, the vehicle behavior, and the reproduction of the environmental and boundary conditions to recreate the potential collision situations (https://youtu.be/AF0nYteBRE8, accessed on 15 August 2022).

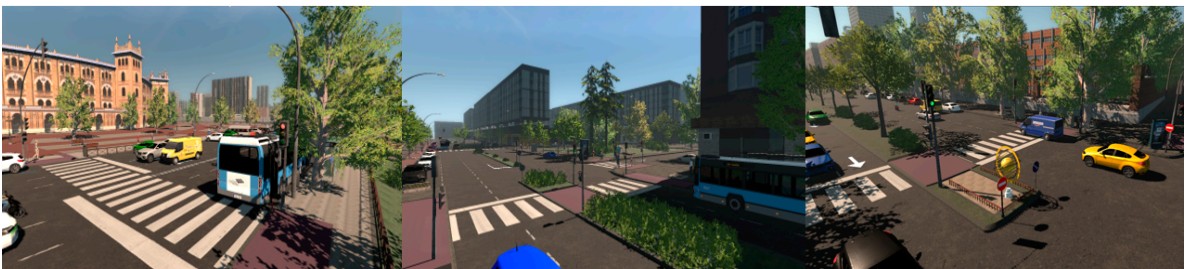

**Figure 2.** A 3D reconstruction of collision scenarios. From left to right: SC1, SC2, and SC3 (Madrid, Spain).

The design process is conducted by merging different data sources, such as online maps, satellite imagery, and photos and videos collected by in situ inspection. The main architectural elements and street furniture are integrated to attain a high level of detail, including buildings, bus shelters, trees, and recycle containers, among others. Close-up photos of the original materials and urban elements have also been used to enhance the user's overall sense of immersion with the interface and generate an environment with a

level of detail close to reality. Additionally, stereophonic sounds have also been included to reproduce the typical background noise in the streets, such as the environmental noise due to motorized traffic and other users interacting on the pavement and the road, as well as the sound of the engine, tires rolling, and hard braking during the accident simulation.

To reproduce the movement of traffic vehicles and pedestrians walking on the sidewalks or crossing the road, the corresponding dynamic elements have been mathematically designed and integrated through a simple kinematic model to replicate their motion and trajectories. Traffic rules have been adapted to emulate the specific operation of traffic lights and flow to allow them to follow the traffic circulation patterns and regulations in force in Spain. Specific modules for modeling environmental conditions (weather and time of day) have been also included to increase the spectrum of traffic conditions to emulate. On the other hand, ad hoc interfaces have been implemented to change the operating conditions of the simulations, including features such as the ratio between the volume of motorized traffic/parked traffic, the percentage of pedestrians walking through the scenario, the operating conditions of the main traffic light (e.g., the waiting time for vehicles, pedestrians, and percentage of blinking), the control of the simulation to trigger the accident event, the environmental conditions, and the type of the attempting vehicle and the main features of its kinematic behavior (propulsion, initial speed, maximum braking deceleration, and driver reaction time).

The simulations and user tests discussed in this paper have been performed using an HP GZ V2 Backpack computer (to run the virtual environment) and an HTC Vive headset. This portable setup allows the user to move across the real environment with a good degree of freedom, improving, therefore, the user's feeling of realism and immersion when interacting with the virtual world.

The tests were carried out at the INSIA facilities, in an enclosed space set up for this purpose, measuring $10 \times 3.2$ m. For each simulation, the application generates a .csv file recording the position of the user's head in Cartesian coordinates and its rotation in Euler angles. The position and speed of the vehicle's center of gravity have also been recorded. The corresponding timestamps are in UNIX format and the interval between two measurements is set to one-tenth of a second. The events related to the simulation of the accident are also registered, such as the start and end of the vehicle's trajectory, the moments when the braking starts and ends, and the activation of the horn and the collision (if any).

*3.3. Development of VR Tests for Pedestrians: Procedure and Questionnaires*

The tests that make up the experimental VR session are held with a sample of 29 users with similar socio-demographic conditions (age: 20–30 years; gender: 28% female, 72% male). All of them are Spanish undergraduate, master's, and doctoral students at the Universidad Politécnica de Madrid (UPM). The gender distribution (Figure 3) is representative of the overall sample of UPM students in undergraduate and postgraduate courses [32].

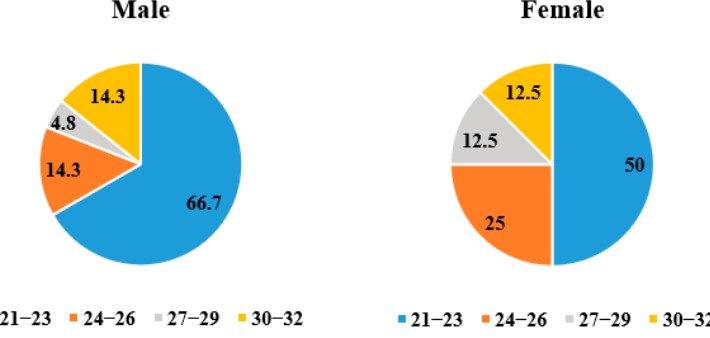

**Figure 3.** Gender distribution of the sample of users that performed the experimental VR session (by age group).

Before starting the tests, each subject fills in a disclaimer with their personal information and history of road traffic accidents, as well as the Simulator Sickness Questionnaire (SSQ), which will be repeated throughout the individual session to identify possible physical affectations in the subject due to VR use. Before the potential collision tests, the System Usability Scale (SUS) questionnaire on the usability of the application is also carried out. Finally, they are also asked to complete the Presence Questionnaire (PQ) for evaluating the interaction with the 3D environment.

Simulator sickness results in a sense of nausea, dizziness, vertigo, and sweating (among other symptoms). To this end, the Simulator Sickness Questionnaire (SSQ) has been used as reported in [6]. In it, the users are asked to rate sixteen possible symptoms on a four-point Likert scale (Table 2). This evaluation allows scoring in the discomfort categories falling including the nausea, oculomotor, and disorientation categories, plus a general score by combining the previous ones.

**Table 2.** Simulation Sickness Questionnaire (SSQ).

| SSQ Items | Nausea | Oculomotor | Disorientation |
|---|---|---|---|
| 1. General discomfort | O | O | |
| 2. Fatigue | | O | |
| 3. Headache | | O | |
| 4. Eyestrain | | O | |
| 5. Difficulty focusing | | O | O |
| 6. Increased salivation | O | | |
| 7. Sweating | O | | |
| 8. Nausea | O | | O |
| 9. Difficulty concentrating | O | O | |
| 10. Fullness of head | | | O |
| 11. Blurred vision | | O | O |
| 12. Dizzy (eyes open) | | | O |
| 13. Dizzy (eyes closed) | | | O |
| 14. Vertigo | | | O |
| 15. Stomach awareness | O | | |
| 16. Burping | O | | |
| *Total* | (1) | (2) | (3) |

| SSQ Components | Computation |
|---|---|
| 1. General discomfort | (1) × 9.54 |
| 2. Fatigue | (2) × 7.58 |
| 3. Headache | (3) × 9.56 |
| *Total score (TS)* | ((1) + (2) + (3)) × 3.74 |

The SUS [33] is a ten-item questionnaire with a five-response Likert scale (Table 3). The Likert formatting ranged from "strongly disagree" to "strongly agree", scaling all values from zero to four, with four being the most positive response (Figure 4). Finally, after summing all individual scores, this result is multiplied by 2.5 to obtain the overall value of SUS. This will convert the range of possible values from zero to one hundred.

The researchers [35] investigated the structure of Version 2.0 of the Presence Questionnaire (32 items) by conducting a cluster analysis of PQ data. Only those PQ items that contributed to the reliability of the scale (19 items) were included in the analysis. The PQ version considered in this paper is a nineteen-item questionnaire with a seven-response Likert scale. The Likert formatting ranged from "Not at all" to "Completely".

**Table 3.** System Usability Scale questionnaire (SUS).

| | Question |
|---|---|
| 1 | I think that I would like to use this product frequently. |
| 2 | I found the product unnecessarily complex |
| 3 | I thought the product was easy to use. |
| 4 | I think that I would need the support of a technical person to be able to use this product. |
| 5 | I found the various functions in the product were well integrated. |
| 6 | I thought there was too much inconsistency in this product. |
| 7 | I imagine that most people would learn to use this product very quickly. |
| 8 | I found the product very awkward to use. |
| 9 | I felt very confident using the product. |
| 10 | I needed to learn a lot of things before I could get going with this product. |

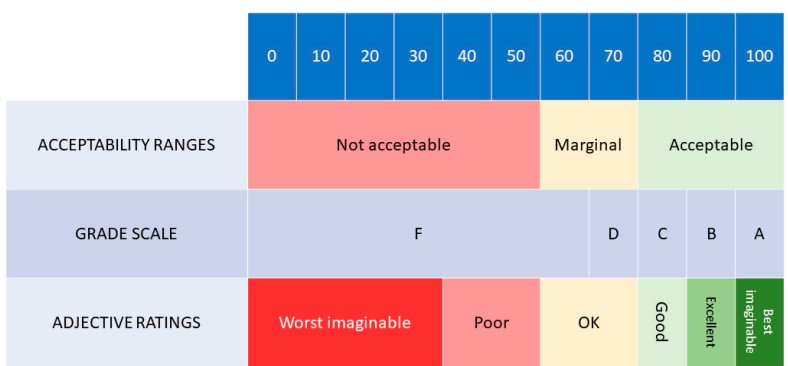

**Figure 4.** A comparison of the adjective ratings, acceptability scores, and school grading scales, in relation to the average SUS score [34].

Each subject must complete the following procedure:

1. **Signing of the participation agreement (personal data disclaimer)**
2. **Completion of SSQ questionnaire (SSQ1)**
3. **Stage 1 (ST1, 10 min).** Familiarization with the VR technology using the Steam VR room. The user is asked to explore the room and recognize the boundaries of the play area, shown virtually through a blue mesh. ST1 lasts 10 min.
4. **Completion of SSQ questionnaire (SSQ2)**
5. **Stage 2 (ST2, 30 min):**
    5.1. *Familiarization with the VR environment in an* ad hoc *training scenario*. The user is asked to walk around to become acquainted with the 3D environment and give feedback on the degree of immersion, clarity, and sound.
    5.2. *Estimation of distances in VR* (cone, 8 m; benches, 10 m; tree, 12.5 m; car, 16 m). The user, from a fixed position in the play area, is asked to estimate the distance a series of objects are in a VR scenario.
    5.3. *Estimation of distances with real objects* (same objects and distances as VR). The user, from the same fixed position, is asked to estimate the distance these real objects are from the same position, without wearing the HMD.
    5.4. *Speed calculation test*. The subject must make three round trips on the pedestrian crossing of a street that is closed to traffic. Thus, the pedestrian's average speed is measured for prospective analysis, and the subject becomes more comfortable walking in the virtual environment (and with the VR headset put on).
    5.5. *Crossing at Toreros Avenue* (SC1). It allows putting the user in a context, simulating real crossing conditions.
6. **Completion of SSQ (SSQ3) and SUS questionnaires.**

7. **Stage 3 (ST3, 15 min):**

  7.1. *Crash tests*. In SC2 and SC3, the pedestrian is requested to cross the road as in real life. When the user is crossing the pedestrian walkway, the application's monitoring team triggers the hit-and-run event, launching a driver-piloted vehicle with no onboard AEB system, that skips the crossing priority.

  7.2. *Calculation of vehicle speed in VR*. The pedestrian, standing on the sidewalk, must estimate the speed of the vehicle crossing the road. The traffic speeds are 30, 40, 60, and 80 km/h, respectively.

  7.3. *Calculation of the safety TTC*. The user is positioned in front of an oncoming vehicle and is asked to react the moment the user considers it unsafe to remain in the position this pedestrian is in. Therefore, considering the gap acceptance and the relative speed at the moment of reaction, the TTC for which the pedestrian considers it safe to react can be worked out.

8. **Completion of SSQ (SSQ4) and PQ questionnaires**

### 3.4. Database Generation

The total sample assessed is 86 tests: 29 tests carried out on Toreros Avenue, 29 tests on Machupichu Avenue, and 28 tests on Hermanos García Noblejas Street. As described above, the Toreros Avenue scenario is part of the familiarization phase. It consists of a test with similar characteristics to the other two scenarios, but with an added pitfall: the vehicle enters the crosswalk from a curved section less than 8 m from the crosswalk, so visibility is more reduced than in the collision scenarios. Nevertheless, this test allows the user to be alerted to the possibility of being run over and to be put on alert for the collision tests so that the prospective response to an analogous event is as realistic as possible.

For each collision scenario, a graph (Figure 5) of pedestrian and vehicle trajectories is conducted, considering their position in the X-Z plane. Subsequently, the points related to the simulation events are indicated: start and end of braking, start and end of the horn, and accident (if any). Similarly, the positions of the pedestrian at the instants corresponding to these events are specified, by taking as reference the timestamp across the registers.

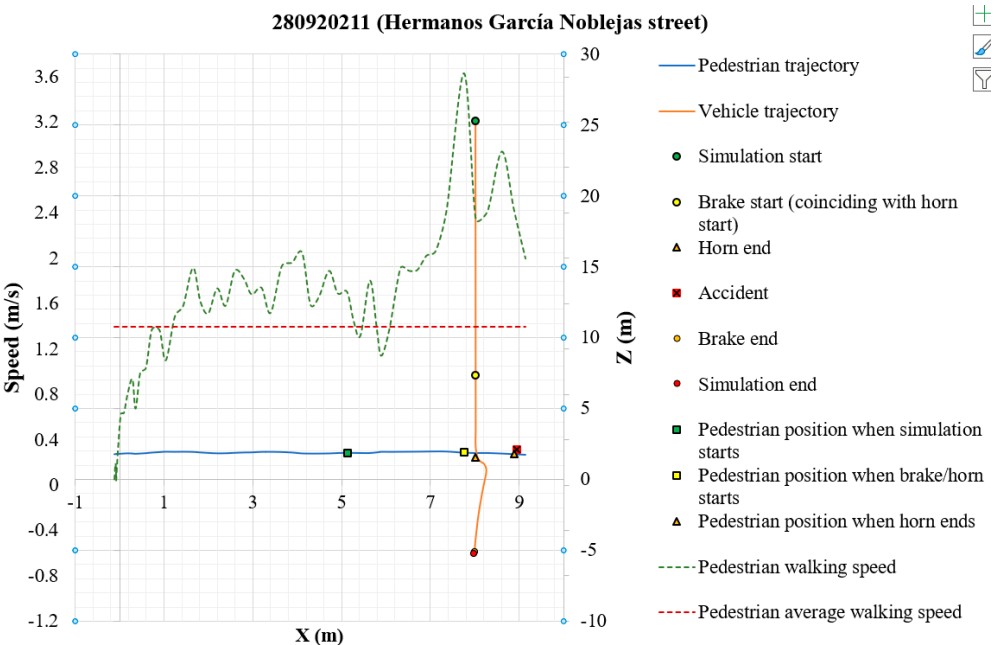

**Figure 5.** Trajectory and speed plots, and positioning of events for a test simulation.

Through the position in X (perpendicular to the road) and Z coordinates (the vehicle is moving parallel to this axis), it is feasible to obtain the instantaneous pedestrian and

vehicle speeds. For this purpose, the distance traveled between two consecutive records is considered, and the time interval being the difference between the corresponding timestamp values is expressed in seconds. Afterward, the pedestrian's speed profile and the evolution of the rotation of the head angle with respect to the *Y*-axis, as a function of the X coordinate, are included in the trajectory plot. Hence, it is attainable to identify both the pedestrian's speed and the direction of the gaze at different moments and positions throughout the simulation. The speed profile is crucial to identify the exact point at which the pedestrian's reaction occurs, if any. The analysis of our data highlighted three typical reaction patterns: accelerating to cross to the curbside at the end of the crosswalk (25.6%); braking and backing up (32.6%); and, finally, not reacting at all (41.8%). Furthermore, the speed profile helps the identification of the moment in which a reaction begins, that is, when there is a monotonous increase (in the case of forwarding acceleration) or decrease (in the case of stopping and backing up) in speed, once the simulation has begun with the vehicle heading towards the crossing.

For this research, to define a parameter to assess the level of attention shown by the user towards vehicles approaching the crossing, the Minimum Attention Angle (MAA) is calculated, as shown in Figure 6. This angle defines a threshold to consider when a pedestrian is estimated to look at the area where a vehicle can potentially appear and enter the pedestrian crossing space. It takes into account the minimum distance the pedestrian must travel to reach a point where the collision is plausible (including the outermost part of the chassis) and the minimum distance the piloted vehicle would need to brake completely from the maximum speed (which is itself the cruising speed), taking as reference the theoretical collision point (the pedestrian walks along the centerline of the crosswalk). Note that in all cases, the vehicle's maximum deceleration has been constant and equal to 7.7 m/s$^2$ ($a_{max}$). If the maximum speed at which the vehicle circulates is 46.9 km/h (13.0 m/s) ($s_{max}$) is considered, the time needed for braking is 1.7 s ($t_{to\ brake}$).

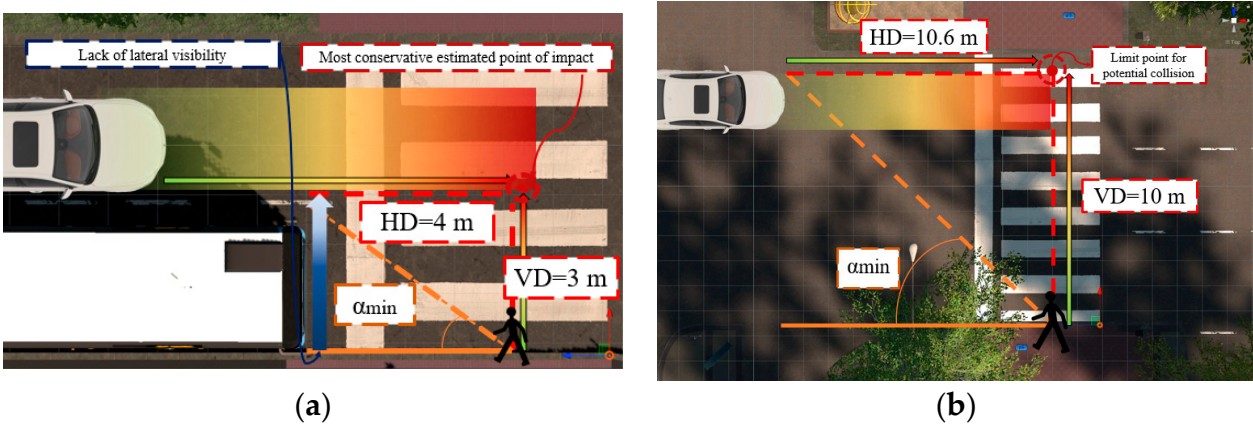

**(a)**          **(b)**

**Figure 6.** Geometric definition of MAA for Av Machupichu (**a**) and Hermanos García Noblejas (**b**).

The minimum distance to accomplish a full stop ($d_{min\ to\ stop}$) is given by the equation:

$$d_{min\ to\ stop} = s_{max} \cdot t_{to\ brake} + \frac{1}{2} \cdot a_{max} \cdot t_{to\ brake}^2 = 10.6\ m \qquad (1)$$

Accordingly, for each scenario, the minimum attention angle (MAA) is defined as:

$$a_{min} = \tan^{-1}(VD/HD) \qquad (2)$$

where VD is the "Vertical Distance" (*X*-axis), and HD (*Z*-axis) is the "Horizontal Distance". For SC2, α = 36.9°, while in SC3 this angle takes the value α = 43.4°.

### 3.5. Data Preprocessing

This research demonstrates how to apply some machine learning classification techniques, including K-Nearest Neighbor (KNN), Support Vector Machines (SVMs), and tree-based models based on Bootstrapping Aggregation methodology, comprising a single Decision Tree (DT) and Random Forest, to determine whether there will be a pedestrian collision based on the degree of attention, the road configuration and the pedestrian's reaction during the crossing.

The data include the individual records for each user who performed the test (57 samples performed in total), thus considering the following features: percentage of time that the pedestrian is looking with a head rotation angle lower than the MAA ("Percentage of attention time", PAT); the reaction type, if any (Reaction type), the type of visibility, complete or reduced, (Visibility); the area where the reaction occurs (Reaction zone), taking as reference the lane in which the vehicle circulates; the average error committed when calculating the distance to real objects (Average error DR); the average error when calculating the distance to VR objects (Average error DVR) (Figure 7); the average error when calculating the vehicle speed in VR (Average error VVR); and the safety TTC. Errors are calculated as (actual value-estimated value)/(actual value).

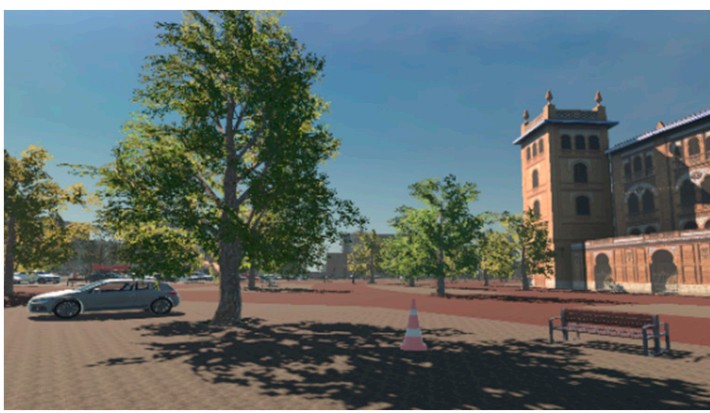

**Figure 7.** VR scenario for estimating distances of different objects: cones, benches, trees, and vehicles.

The preprocessing of the variables was conducted with Python and SPSS®, while the model development was performed entirely in Python. The PAT variable, the TTC, and the errors in the calculation of distances and speeds are numerical variables, while the remaining are categorical variables. Similarly, the response variable "Event" is a discrete variable. Therefore, categorical variables must be transformed into numerical values to be analyzed by classification techniques, especially in the case of distance-based models. For the explanatory variables, the One-Hot Encoder (OHE) function has been used, while for the target value the LabelEncoder function has been applied, both contained within the Python module sklearn.preprocessing. The OHE function generates a binary column for each category, returning a sparse matrix, while LabelEncoder encodes the response variable by assigning values between 0 and nclasses-1, so that "0" is assigned for the label "Avoidance" and "1" for "Collision". Within the OHE function, it is specified that one of the categories should be dropped if the variable is binary, to reduce to some extent the effects of multicollinearity.

For the distance-based classifiers, Feature Scaling is applied to the input variables, since the range of values of the variable "PAT" is different from the one considered in the new binary variables obtained from the encoding. For this purpose, a standardization of the variables is conducted to eliminate the mean and scale them to unit variance. By normalizing the features, it is avoided that those variables such as "PAT", which have a variance whose order of magnitude is greater than the rest, could be dominant in the objective function and cause the classifier not to learn correctly from the rest of the variables.

To determine whether two variables are related, some measures of association with their corresponding significance test must be used [36]. The succeeding bivariate analysis and graphics have been implemented in SPSS® software. The method used for feature selection is derived from the minimum redundancy maximum relevance (mRMR) approach [37], consisting in selecting those features that are more correlated to the response variable and which are less correlated with each other.

To determine the correlation statistic to be used between the numerical explanatory variables, the Kolmogorov–Smirnov (K–S) test with Lilliefors correction was performed. This non-parametric test shows that there is no normality for the variables "PAT", "TTC" and "Average error VVR", but there is normality for the variables "Average error DR" and "Average error DVR" (Table 4). Therefore, only Pearson's correlation will be used to estimate the correlation between variables that result in a normal distribution from the K–S test, while the correlation between variables with normal distribution and those significant in the K–S test will be measured with Spearman's correlation coefficient (Table 5).

**Table 4.** Significance level of the Kolmogorov–Smirnov test for each numerical explaining variable.

| PAT | TTC | Average Error DR | Average Error DVR | Average Error VVR |
|---|---|---|---|---|
| <0.001 | 0.0388 | 0.2 | 0.062 | 0.031 |

**Table 5.** Correlation values for continuous explaining variables.

| | PAT | TTC | Average Error DR | Average Error DVR | Average Error VVR |
|---|---|---|---|---|---|
| **PAT** | 1 | 0.128 (sig = 0.343) | −0.106 (sig = 0.433) | −0.117 (sig = 0.388) | −0.141 (sig = 0.295) |
| **TTC** | 0.128 (sig = 0.343) | 1 | −0.329 (sig = 0.012) | −0.355 (sig = 0.007) | 0.112 (sig = 0.409) |
| **Average error DR** | −0.106 (sig = 0.433) | −0.329 (sig = 0.012) | 1 | 0.498 (sig = 0.001) * | 0.026 (sig = 0.848) |
| **Average error DVR** | −0.117 (sig = 0.388) | −0.355 (sig = 0.007) | 0.498 (sig = 0.001) * | 1 | −0.077 (sig = 0.571) |
| **Average error VVR** | −0.141 (sig = 0.295) | 0.112 (sig = 0.409) | −0.026 (sig = 0.848) | −0.077 (sig = 0.571) | 1 |

* To measure the correlation between these explaining variables, the Pearson coefficient was used; in the remaining, Spearman rho was used.

Since not all numerical explanatory variables fit a normal distribution, assessing the level of correlation with the variable "Event" through the point biserial correlation coefficient is discarded. Therefore, binary logistic regression would allow for evaluating the level of association of the numerical predictor variables with the response variable. Thus, requirements such as linearity, normality, or homoscedasticity should not be tested. Considering all continuous explanatory variables, the Wald elimination method is carried backward to leave in the model those variables that significantly affect the dichotomous response variable. The variables with the highest significance on the variable "Event" are "PAT" (sig = 0.002) and "Average error DVR" (sig = 0.03).

Likewise, the results in (Table 5) justify the independence between these two predictors, so that "PAT" and "Average error DVR" will be included in the classification models to be addressed in the following sections.

In the case of categorical explanatory variables, the Chi-Square test allows us to evaluate whether the variables are independent of each other and of the response variable. In this case, the null hypothesis (H0) is that the variables are independent, while the alternative hypothesis (H1) is that there is a relationship between this pair of variables. It is not necessary in any case to use Fisher's exact test since the expected counts less than five are less than 20% in all cases (Table 6).

**Table 6.** Chi-square test results for categorical explaining variables and the response variable.

|  | Reaction Type | Reaction Zone | Street Type | Event |
|---|---|---|---|---|
| **Reaction type** | 1 | 60.034 (sig < 0.001) | 0.541 (sig = 0.763) | 41.464 (sig < 0.001) |
| **Reaction zone** | 60.034 (sig < 0.001) | 1 | 7.738 (sig = 0.021) | 36.371 (sig < 0.001) |
| **Street type** | 0.541 (sig = 0.763) | 7.738 (sig = 0.021) | 1 | 0.153 (sig = 0.696) |
| **Event** | 41.464 (sig < 0.001) | 36.371 (sig < 0.001) | 0.153 (sig = 0.696) | 1 |

Reaction and type of reaction are strongly related to the response variable, while the street type variable is not significant. Although the reaction and reaction type variables are correlated, it was decided that both predictors should be part of the model since, if one of them were eliminated, relevant information would be lost to characterize pedestrian behavior (especially in the decision tree models).

Therefore, the variables added to the models are Reaction type, Reaction zone, PAT, and Average error DVR.

From the clustered bar charts (Figure 8), the following inferences can be drawn:

1. The probability of being run over is significantly lower when the pedestrian stops and backs up than when accelerating, where the distribution of cases is 58.3% for accidents and 41.6% for avoidance. Failure to react is irrefutably implicated in a hit-and-run accident
2. Reacting before reaching the lane in which the vehicle is traveling means an avoidance rate of 85.7%, while reacting once in that lane means being hit is 57.1%. In case of not changing the walking speed, the collision is ensured.

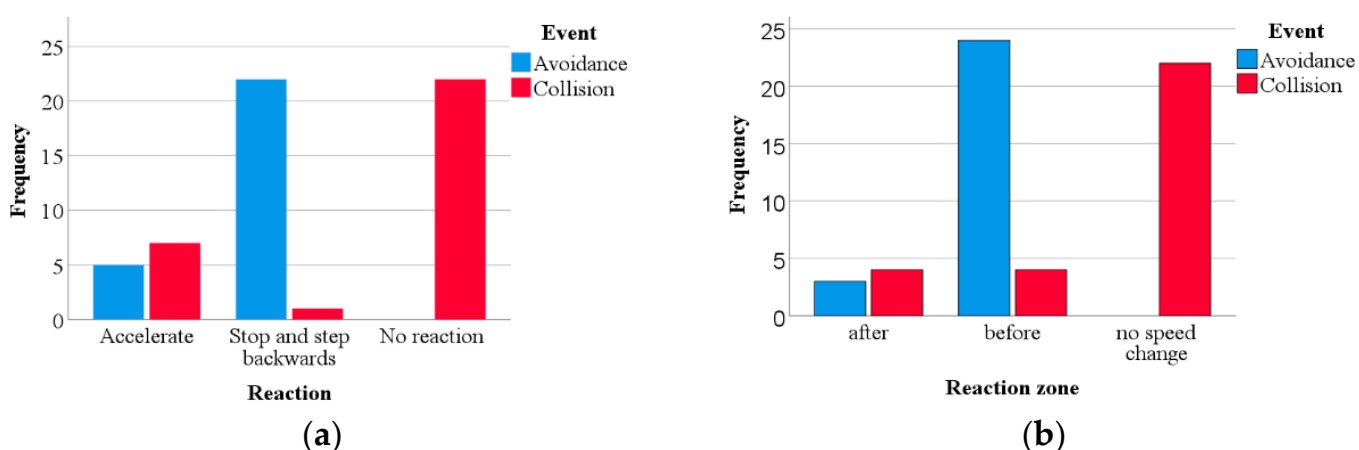

(**a**)  (**b**)

**Figure 8.** Clustered bar charts of predictors "Reaction" (**a**) and "Reaction zone" (**b**).

In Figure 9, the cases with a greater miscalculation in distances during the experimental session in VR tend not to react to potential run-over situations (Q1 higher than the interquartile range of the other two types of reaction). This is evidence that, when perceiving the vehicle closer than it really is, the pedestrian tends to remain blocked or paralyzed, thus nullifying their response capacity.

In the scatter plot in Figure 10, notice how there is a great accumulation of avoidance cases for higher percentages of time with greater attention, while those who spend less time looking at an angle less than that defined as the limit tend to be hit. Some cases contradict this hypothesis, so they cannot be explained without the remaining estimators.

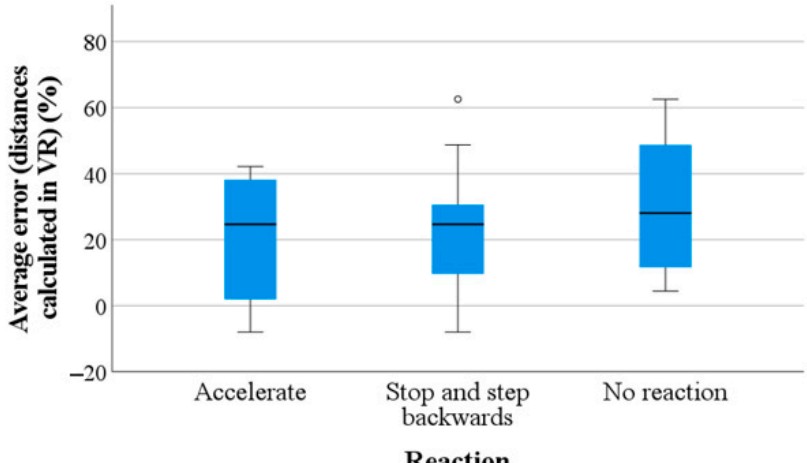

**Figure 9.** Boxplot chart of "Average error of distances calculated in VR" filtered by Event.

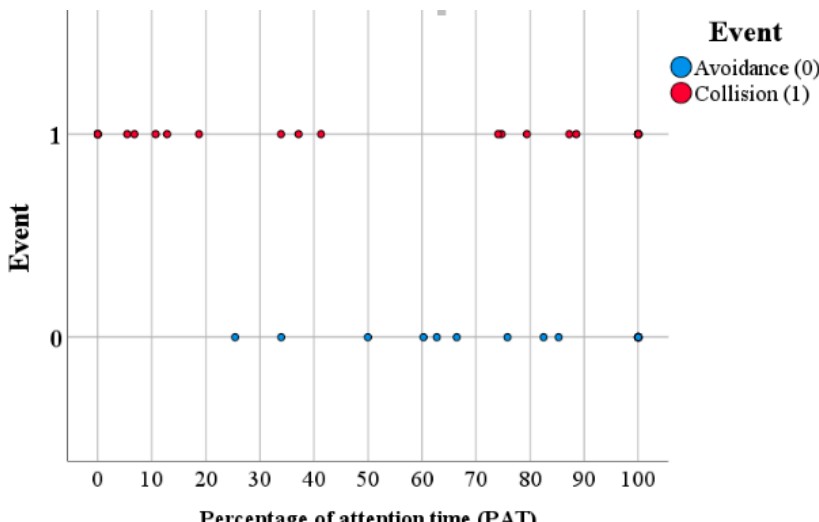

**Figure 10.** Grouped Scatter of "Event" by "Percentage of attention time".

## 4. Results

### 4.1. Questionnaires Results

The SSQ questionnaire was filled at four different moments during the experimental sessions, as described in the previous section. The initial response (TS1) was subtracted from each of the three later responses to obtain the change in total sickness measure (ΔTS2, ΔTS3, and ΔTS4) as metrics to determine discomfort on the HMD (Figure 11).

This figure shows the distribution for each of the three metrics considered. The results show a high dispersion. In 45% of the users, there was no change in the Total Score during the performance of the trials (ΔTS2 to ΔTS4 are equal to zero). The percentage of ΔTS scores greater than 10 increases from stage 1 (ΔTS2) to stage 3 (ΔTS4), indicating users experienced a greater increase in discomfort during this stage (ΔTS4).

The SUS provides a broad global view of subjective assessments of usability. The SUS was administered to 29 participants. The results show a high mean SUS score across all participants at 82.9 (SD 11.6). Eighty-three percent of the participants assess the VR system as "Acceptable" and seventeen percent as "Marginal" (Figure 12), according to the acceptability scores shown in Figure 4. There is no user who rated the system as "Not acceptable" (SUS less than 50).

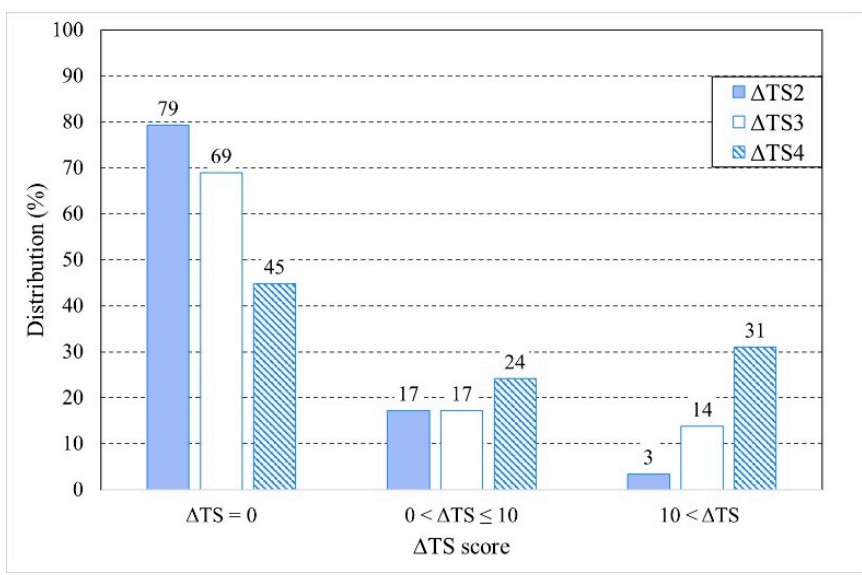

**Figure 11.** Change in total sickness measure of each stage.

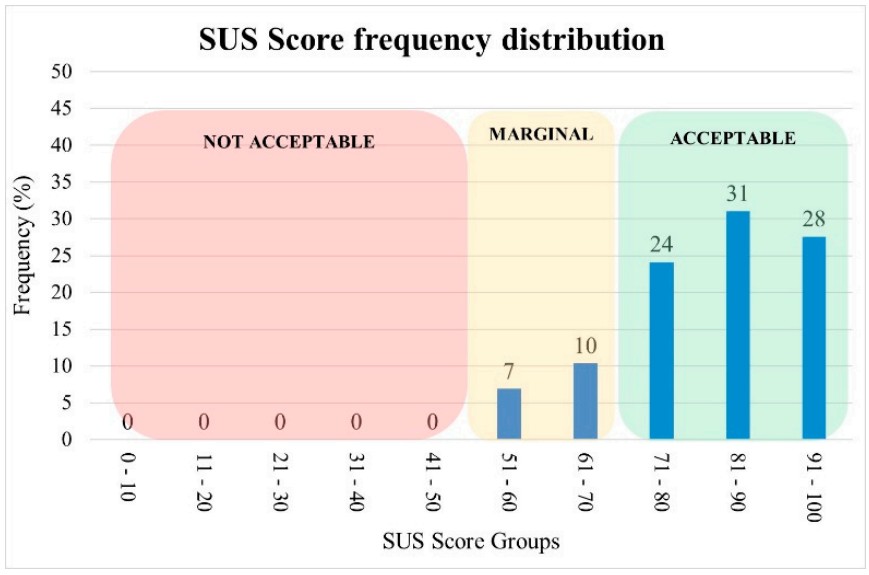

**Figure 12.** SUS score frequency distribution across all participants.

The PQ was administered to 29 participants and the results show a high mean PQ score across all participants at 5.3 (SD 0.3).

### 4.2. Model Fitting

Classification algorithms are supervised learning methods that are widely used to solve predictive modeling problems when the response variable is labeled into classes or categories. Since the response variable considered is dichotomous, this is a binary classification problem (Collision: "1"; Avoidance:"0").

The accuracy and performance metrics for each of these methods have been calculated using the k-fold cross-validation technique [38], which allows different partitions of data to be selected between the training set and test set, thus ensuring the generation of independent data partitions. By using a total of k = 5 iterations, the possible overfitting derived from a small training data size is avoided, thus obtaining a higher reproducibility in the whole sample and higher reliability in obtaining the accuracy through the arithmetic mean of all k repetitions.

To obtain the hyperparameters that optimize the overall performance of the models, the GridSearchCV function of the sklearnmodel_selection module is used. Table 7 lists these hyperparameters, which will be described in the following paragraphs.

**Table 7.** Model hyperparameters and their range of values.

| Model | Hyperparameters |
|---|---|
| **SVM** | *C*: [1; 10; 100; 1000]; *Kernel function*: [rbf; linear]; *gamma* (*only for rbf*): [$1 \times 10^{-3}$; $1 \times 10^{-4}$] |
| **KNN** | *K*: [1–31]; *metric* (*p*): [1, Manhattan; 2, Minkowski] |
| **Individual decision tree \*** | *Split criteria*: [Cross-Entropy;Gini-index] |
| **Random Forest \*** | *Split criteria*: [Cross-Entropy;Gini-index], *number trees*: [determined through Out-of-the-bag error criteria] |

\* *Minimum sample splits* were left to 2 and *minimum sample leaf* was left to 1.

The distance-based models are:

(1) *Support Vector Machine (SVM):* This method is based on the construction of a hyperplane or set of hyperplanes in a space of very high (or even infinite) dimensionality to achieve a good separation of classes to obtain a good classification. Therefore, the greater the distance of the hyperplane to the training data points that are closest to each other, the smaller the error made by the classifier [39].

The mathematical functions used in the SVM algorithm, known as Kernel functions, can be linear, polynomial, radial basis function (RBF), sigmoid, etc. These functions assign the data to a different, generally higher dimensional space, intending to separate the classes more simply once the transformation has been carried out and thus simplifying the complex and non-linear decision limits to make them linear in the primitive high-dimensional space [40].

Since the study does not work with linearly separable data, the objective function consists of minimizing the first term, equivalent to maximizing the margin between classes, and a second term, where the error associated with the margin violation is multiplied by a regularization term C. Ideally, the term would be greater than or equal to one, indicating perfect accuracy. However, the introduction of the penalty term C allows controlling the strength of the penalty incurred when a sample is misclassified or within the margin boundary. It is therefore obtained that the highest accuracy values are achieved with a Radial Basis Function (RBF) kernel, Gamma: $1 \times 10^{-3}$, and C = 1000. Note that the value of the regularization parameter C is high enough to state that a lower margin will be accepted if the decision function is better at classifying all the points of the training set correctly. Nevertheless, this high value does not condition the accuracy obtained (86.1%).

(2) *K-nearest neighbors (KNN):* It is a classification method that enables us to estimate the density function of the predictors or observations $X_i$ for each class $C_j$ of the response variable or directly the a posteriori probability that this element belongs to the corresponding class. This is a lazy type of learning since the function is approximated only locally, computing from a simple majority vote of the K nearest neighbors of each observation [41]. This is a robust method when there is noise in the training dataset, especially when the value of K is large in that case. Similarly, if the value of K is high, it can create boundaries between similar classes.

In this model, the distance metric used is Minkowski, so the distances are calculated with the standard Euclidean metric. Applying the GridSearchCV optimization function, the highest accuracy is obtained for K = 8 (91.5%).

On the other hand, the tree-based models considered are:

(3) *Individual decision tree:* This is a non-parametric prediction model based on the construction of diagrams with a tree-like structure [42]. The decision tree divides the sample space into subsets through a series of decision rules by means of a recursive (top-down)

partitioning, so that the process ends when the subset at a node has all the same value as the target variable, or when the partition no longer adds value to the predictions.

Different criteria can be used to select the variable and split it successively, although the most usual ones are the Cross-Entropy and Gini index [43]. Both are based on the probability of each class, adding the logarithmic term in the case of entropy calculation, which makes this criterion computationally more complex. Therefore, and given that both methods yield similar results, the Gini index has been chosen for this study.

The decision tree was imported from the sklearn.tree module of the Scikit learn Python library. As can be seen in Figure 13, the first filter on the starting training sample is the type of reaction. In cases where the pedestrian stops and backs up, the accident is avoided. In case of reacting differently (or not reacting at all), any situation that involves not performing such action before the hit lane will imply a collision. In cases where the pedestrian accelerates early before the lane in which the vehicle is traveling, there is a PAT value below 33.9%, since the vehicle starts the simulation implies an accident. If PAT is higher than 33.9%, only in cases with higher visual acuity in the estimation of distances in VR (average error lower than 31.1%), the accident is avoided, while for average error values higher than 31.1%, the accident is avoided with a PAT lower than 67%. The accuracy obtained is 89.85% after k-fold cross-validation is implemented.

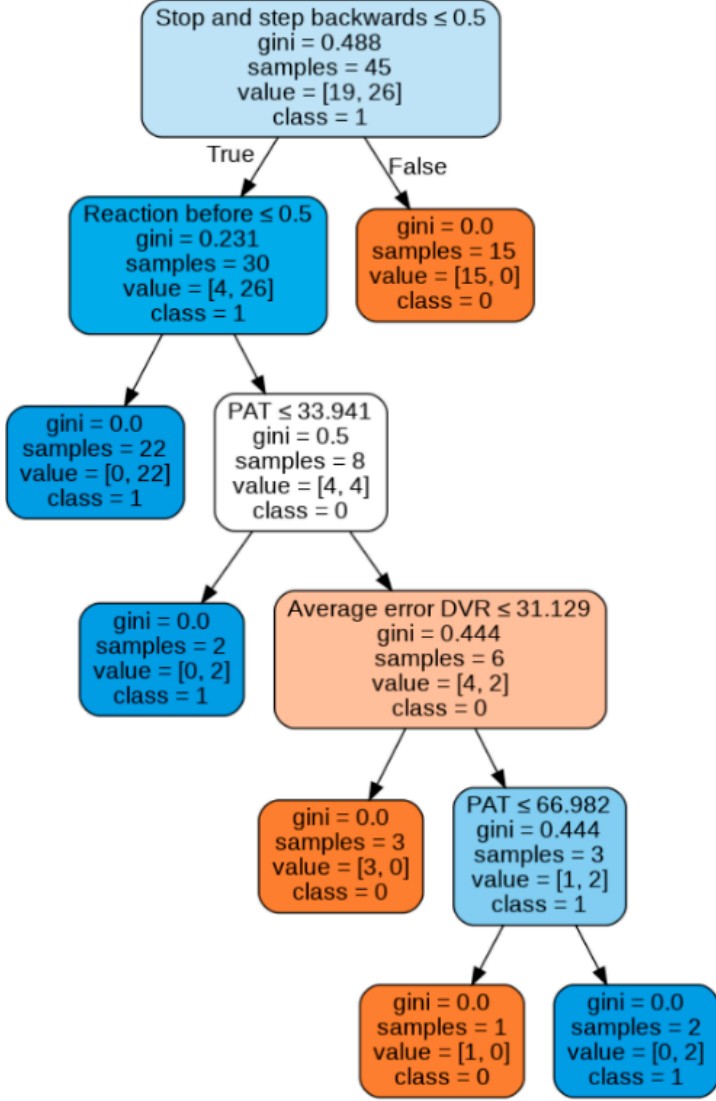

**Figure 13.** Individual decision tree model.

(4) *Random Forest:* This method [44] consists of an ensemble of uncorrelated decision trees, the result of which is obtained through the majority vote criterion. The algorithm is based on the bootstrap aggregating or bagging technique, which is based on the random selection of a subsample with a replacement of the training set to feed and fit each of the trees. The advantage of this procedure is better model performance because it reduces the variance of the model, without increasing the bias, and thus reduces sensitivity to the noise that can be obtained with a single tree. In this technique, adding more trees to the model does not increase the risk of over-fitting, but beyond a certain number of trees, no benefit will be achieved. To determine the optimal number of trees, the out-of-bag (OOB) error rate is used as a criterion. The OOB error [45] is the mean error for each $y_i$ value (calculated using predictions of the trees) that does not contain this observation in its corresponding Bootstrap sample.

This parameter is calculated by reusing the existing fitted model attributes to initialize the new model in the subsequent call to fit. In this case, this methodology is used to build Random Forest and add more trees, but not to reduce the number of trees. It can be seen in Figure 14 that, from 24 trees, the OOB error stabilizes and adopts a constant value. Therefore, for a number of trees equal to this value, the Random Forest behaves stably.

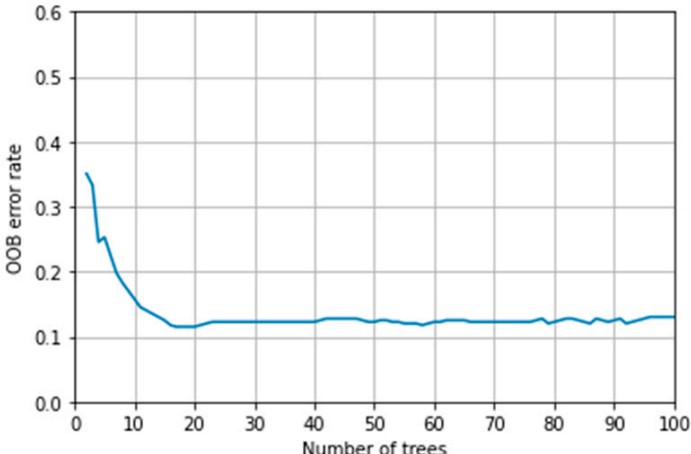

**Figure 14.** Out-of-bag (OOB) error rate vs. number of trees.

In the example tree shown in Figure 15, the number of values in the sample does not coincide with the sum of the values of each class. This is due to the implementation of the bootstrapping metalgorithm, which generates new subsets by uniform sampling and replacement over the original dataset, which means that during sample extraction one of the observations can repeatedly come into other samples when fitting the model.

The trees discriminate consistently on the first branches, identifying that reacting once entering the hit-and-run lane ensures collision. As for the individual decision tree, braking is equivalent to avoiding a collision. Complementarily, not reacting or not modifying the walking speed triggers the hit-and-run. For any other reaction (i.e., accelerating) before the run-over lane, a collision is avoided when the average error in the VR distance calculation is lower than 31.1% and the PAT higher than 42%, while for cases where the error is higher than 31%, a collision is avoided only when the PAT is lower than 67%. This method achieves a final accuracy of 84.6%.

The first limitation of both tree models is to assume that in all cases in which the pedestrian stops and backs up, a collision is avoided. However, the error made in this assumption is minimal since this assumption occurs in 96% of the cases (only 1 outlier). The second limitation of these approaches is the lack of logical reasoning in the last node of both models, since avoidance is assured for intermediate values of PAT. However, it hits the target value of two outliers, where an accident occurred even when looking with a sufficiently high PAT, increasing the final accuracy obtained.

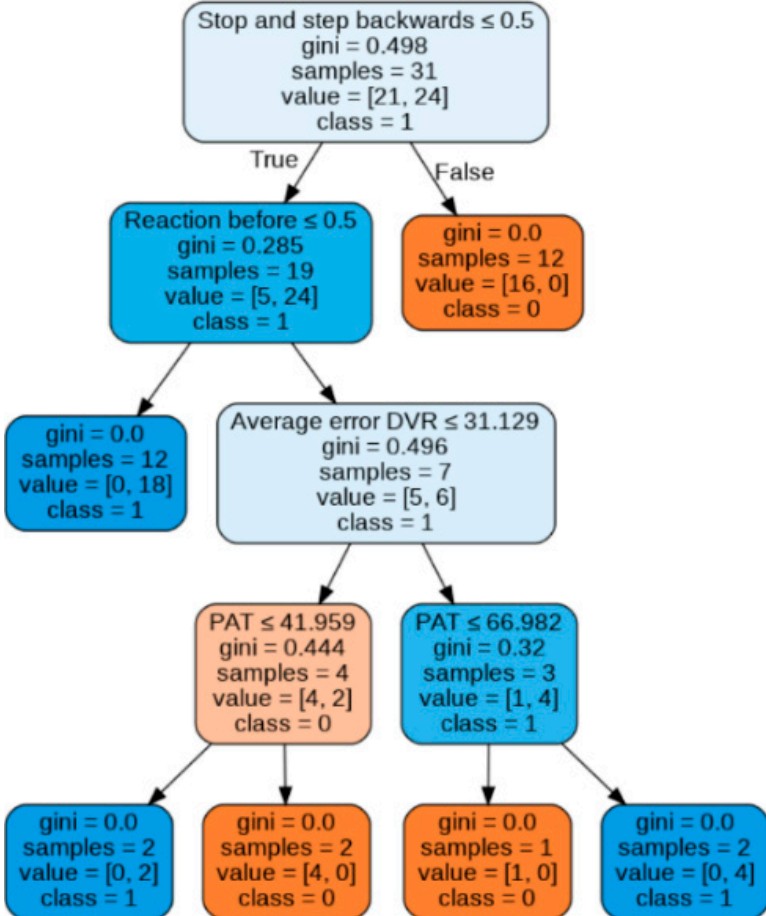

**Figure 15.** Individual tree of the Random Forest ensemble.

The importance of the features [38] introduced to the model has been determined using two measures: Mean Decrease in Impurity (MDI), and Mean Decrease Accuracy (MDA). The MDI importance metric describes how each individual feature contributes to the decrease in the Gini index across trees, while MDA measurement is defined as the decrease in a model score when a single attribute value is randomly shuffled (permutation). Since the Gini importance measurement may be conditioned by the low cardinality of the features or by those categorical variables with a small number of classes, the evaluation of permutation importance is preferred. For this analysis, a pipeline is built that includes a processing stage for the encoding of the variables and a classifier execution stage, so that the encoding stage can be accessed for the impurity measurement.

Reaction stopping and backtracking is the most important variable following the MDI metric, followed by reaction before the run-over lane (Figure 16). In particular, it is possible to see how the impurity-based feature importance inflates the importance of the numerical features "Average error DVR" and "PAT." This occurs because one of the limitations of the MDI metric is that the calculated importance is biased towards high cardinality features. On the other hand, using the permutation criterion, it is noticeable (Figure 17) that the reaction type remains the most important feature for the development of the model, followed by the reaction zone, while the numerical variables reach a lower value of importance in the ranking of predictors, as it is reflected in the splitting order of the nodes in both tree models.

Finally, Table 8 summarizes the optimized parameters that were chosen for each model, corresponding to the previous explanation:

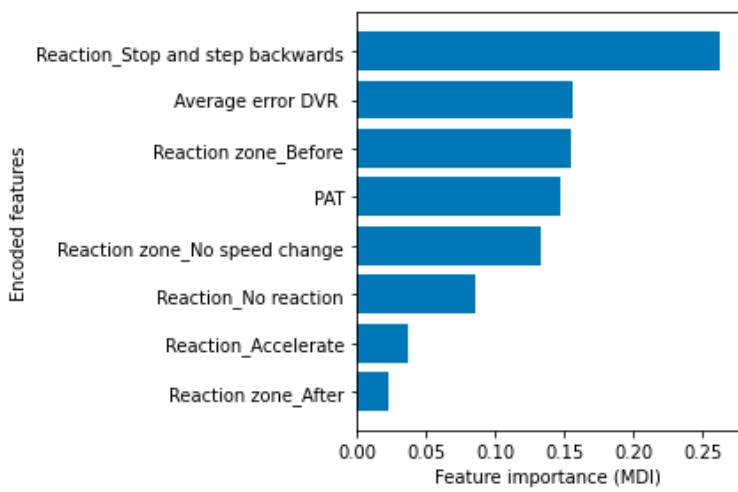

**Figure 16.** Random Forest feature importance using MDI criterion.

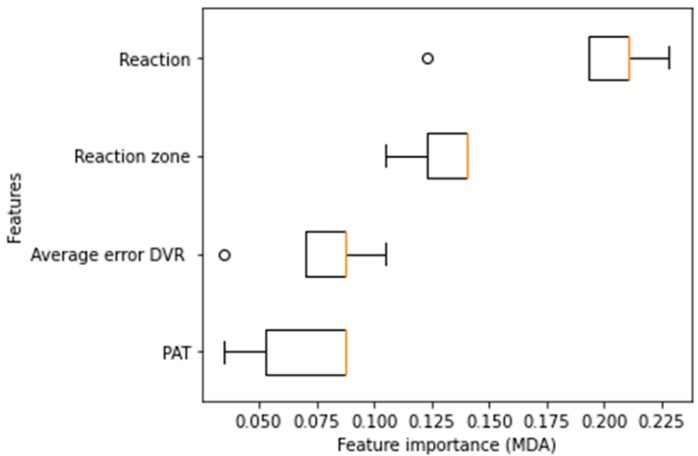

**Figure 17.** Random Forest feature importance using MDA criterion.

**Table 8.** Optimized hyperparameters.

| Model | Hyperparameters |
|---|---|
| **SVM** | *C*: 1000; *Kernel function*: rbf; *gamma (only for rbf)*: $1 \times 10^{-3}$ |
| **KNN** | *K*: [1,31]; *metric (p)*: 2 (Minkowski) |
| **Individual decision tree \*** | *Split criteria*: Gini-index |
| **Random Forest \*** | *Split criteria*: Gini-index, *number trees*: 24 (OOB rate stable around 0.12) |

\* *Minimum sample splits* were left to two and *minimum sample leaf* was left to one.

## 5. Discussion

In this section, to evaluate the suitability of the previously analyzed classification models, overall accuracy is not the only metric to be considered. Although it is not an imbalanced sample, it is useful to measure the success of prediction in each of the classes (collision and avoidance cases) through a set of performance metrics (especially precision and recall). Finally, the analysis of the performance of these classifiers is also completed by means of the Receiver Operating Characteristic (ROC) curves.

Mathematically, precision is defined as the ratio of true positives to the sum of true positives plus false positives, while recall (or sensitivity) is the result of dividing true positives by the sum of true positives and false negatives. In other words, precision

measures the ability of the classifier to identify the relevant data points and the recall to find all the relevant cases within the sample.

From the point of view of future implementation in the decision algorithm of an AEB system, it is primary to measure the precision and recall for the "Collision" class. In the case of obtaining a very high recall, it would result in a lower precision. That is, the model would label a large proportion of trials as collisions, even when the event resulted in avoidance. If such a model were included in the algorithm of an AEB device, there would be a risk that the system would act conservatively by executing a more determined and abrupt braking maneuver in those cases where it is not necessary, generating avoidable wear on the braking system and reducing the risk of causing a rear-end collision by taking a more anticipatory action. Similarly, a very high precision would classify many cases as "Avoidance" and not as an accident, which would imply that the system would not act promptly in risky cases. Therefore, to avoid these two extreme scenarios, a trade-off between these two metrics should be maximized. An optimal blend of precision and recall is the F1-score, defined as the harmonic mean of both, giving equal weight to each.

Of the four models, the ones with the most balanced precision and recall values are tree models and SVM (Figure 18). On the contrary, the KNN has lopsided values for these two metrics, considering that the precision for the collision cases is notably higher than the corresponding recall, which would imply a lack in the detection of a possible accident. The SVM tends to behave similarly in terms of these metrics and the F1-score, although, given the overall accuracy value (Table 9) of the individual decision tree, this is more recommendable.

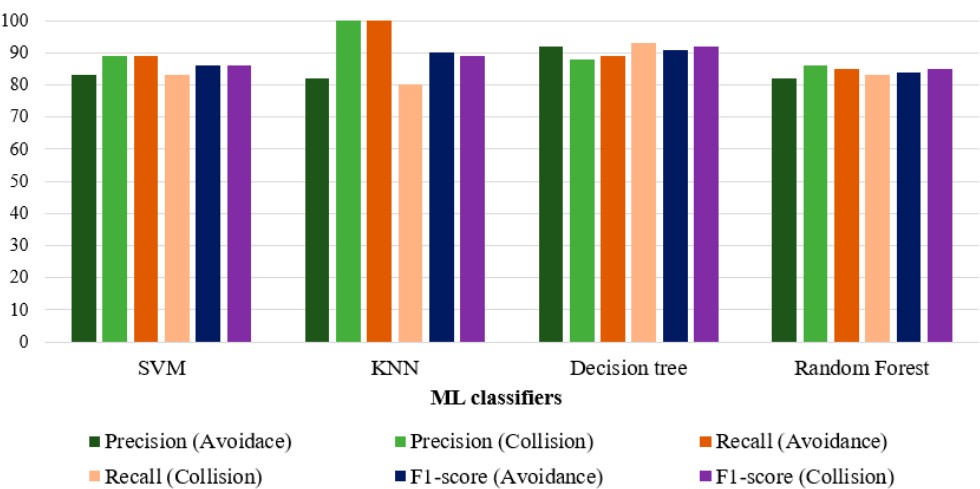

**Figure 18.** Precision, recall, F1-score, and accuracy values for each classifier model.

**Table 9.** Accuracy values of Machine Learning classifiers.

|  | SVM | KNN | Decision Tree | Random Forest |
|---|---|---|---|---|
| **Accuracy** | 86.1% | 89% | 91.5% | 84.6% |
| **Precision** | 86% | 91% | 90% | 84% |
| **Recall** | 86% | 90% | 91% | 84% |

The results obtained can also be illustrated through the ROC curves in Figure 19. These curves show the trade-off between the True Positive Rate (sensitivity) and False Positive Rate (1-specificity). Note how in all models the average area under the curve (AUC) is higher than 0.95 applying cross-validation with k = 5. Likewise, the tree-based models present ROC curves farther away from the line of no-discrimination, resulting in values closer to perfect classification when varying the threshold used to predict the classes.

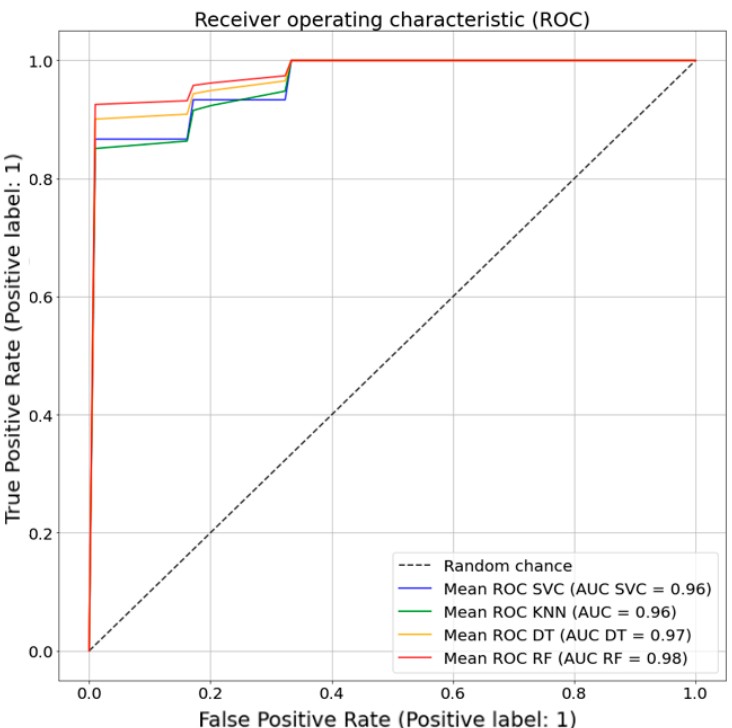

**Figure 19.** ROC curves and AUC value of each classifier.

## 6. Conclusions

In this research, different approaches to classification models have been evaluated to determine the existence of a potential collision, given a series of factors that define pedestrian crossing behavior. These predictors have been extracted and calculated from data obtained from tests in urban scenarios in VR where users had to face potential hit-and-run situations. The integration of VR technology in the assessment of pedestrian behavior's main patterns when crossing an urban road is a worthwhile method. To this end, the movement of vehicles and pedestrians walking on sidewalks or crossing the road, the main architectural elements and street furniture are integrated to attain a high level of detail. This realistic environment has allowed numerous pedestrian tests to be carried out under safe and repeatable conditions.

The questionnaires used during the VR tests show that users experienced a greater increase in discomfort after 55 min. Most of the participants assessed the VR system usability as "acceptable" and they evaluated the effect of immersion as "consistent".

For the KNN and SVM models, an optimization function has been implemented to determine the hyperparameters that maximize their accuracy. For the Random Forest ensembled tree model, the OOB error criterion was used, and the importance of the variables was additionally measured.

From the results obtained, it is concluded that all the models offer an acceptable accuracy, higher than 80%. If the labeling performance of each class is considered, the tree models offer more balanced values in terms of accuracy and sensitivity. However, despite achieving high accuracy in all the classification models, it is not enough to make the decision algorithm of the ADAS system depend solely and exclusively on the output generated by these classification approaches, but it would complement the algorithm since the result obtained in real-time from these models can be used to regulate the braking response of the AEB itself. Thus, in cases where collision avoidance is predicted, the braking pressure can be a fraction of the maximum pressure, or the braking progressivity can be increased, delaying in time the point of maximum deceleration. Therefore, the ML model included in the AEB is complementary and never a substitute, always guaranteeing braking safely in a potential collision situation.

Another noteworthy and differentiating aspect is the internal structure of the models. Both SVM and KNN are "black-box" models, while tree models can be categorized as "white-box" [46]. Commonly, black-box models are those containing complex mathematical functions (SVM) and those requiring a deep understanding of the distance function and spatial representation (KNN). White-box models are preferred for practical application since the decision algorithm is very similar to that of human language. Given the characteristics of the problem, the accuracy obtained and that a model where it is possible to identify in a less straightforward way how the algorithm behaves and how it should be tuned is more profitable for future implementation in a logic system of an AEB device, the individual decision tree is the most preferable method for this purpose.

As a reference, [47] proposes a similar approach, also based on variables related to the pedestrian's movement and posture, captured by an onboard front camera. In this case, the response variable is based on the hazard determination (safe, danger), and the judgment is equivalent. The results obtained in terms of accuracy and recall for the technique used (in this case, fuzzy rules) are 94% and 87%, respectively. Compared to the results obtained in this research, the ML methods are used to achieve a higher trade-off between precision and recall, achieving a similar final accuracy in the case of the individual decision tree.

Since the pedestrian's level of attention when crossing the road is a determining factor in the prediction of collisions, the AEB system must be able to recognize those moments when the pedestrian is looking at the approach zone of the vehicle. For this purpose, the design of a face and eyes recognition system is being developed, consisting of a small single-board computer (SBC) and a high-quality resolution camera with a varifocal lens. Eyes and face landmarks detections are achieved through Open CV (Python) and MediaPipe. The system is intended to identify if the pedestrian is looking at the area the vehicle is approaching (following PAT thresholds criterion) as a function of the pedestrian's interpupillary distance and the distance between the pedestrian and the vehicle.

In practice, the remaining predictors included in the model can be detected by the current ADAS systems integrated into current vehicles. The type of reaction implies a change in the pedestrian's kinematics, the speed being captured by the fusion sensor itself. The camera, in conjunction with the lane detection system, would allow us to calculate the relative position of the pedestrian in the lanes of the roadway.

Regarding the limitations of this research, the simplicity of pedestrian pattern recognition can not only be useful for the technological development of vehicle safety appliances, but also in comprehending pedestrian behavior under certain conditions, and how particular cases escape logical reasoning, breaking the assumptions of the models and negatively affect its predictions. To improve the outcomes of our methodology, an important step will be to increase the sample size, so as to have more data to work with Additionally, we are working on including information about cyclists and considering the information coming from eye-tracking systems. On the other hand, the average error in the calculation of distances in VR is information that must be accessible through V2P (Vehicle-to-Pedestrian) technology, which allows cloud communication between the automobile and the vulnerable user. In this way, the vehicle interface will be able to obtain direct information from this data when connected to a smart device. The limitations of this technology may be the speed of data transfer and the extension of the VR distance estimation experiment to the population. Moreover, as part of the future lines of research planned with VRU, experimental VR tests with cyclists will also allow determining the difference in behavior between the different VRUs and how they interact under different hit-and-run scenarios, thus obtaining more complete and thorough models. As for the equipment, in addition to the portable backpack, it includes a mountain bike, whose rear wheel is placed on a training roller, which acts as a resistance to movement. A controller must be located on the handlebars to track the corresponding turn, and a dongle allows a receiver located on the rear wheel to be wirelessly connected to the computer. Parameters such as turning speed, pedaling frequency, or handlebar turning angle will be used to model the cyclist's behavior and reaction in the phase prior to a potential collision.

**Author Contributions:** Conceptualization, Á.L., F.J.P., F.L. and L.P.; methodology, Á.L., F.J.P. and L.P.; software, Á.L., F.J.P., F.L. and L.P.; validation, Á.L., F.J.P. and L.P.; formal analysis, Á.L.; investigation, Á.L., F.J.P., F.L. and L.P.; resources, F.J.P., F.L. and L.P.; data curation, Á.L.; writing—original draft preparation, Á.L. and F.J.P.; writing—review and editing, Á.L., F.J.P., F.L. and L.P.; visualization, Á.L., F.J.P., F.L. and L.P.; supervision, Á.L. and F.J.P.; project administration, F.J.P.; funding acquisition, F.J.P. All authors have read and agreed to the published version of the manuscript.

**Funding:** This research was funded by the Project OPREVU Grant RTI2018-096617-B-100 funded by MCI/AEI/10.13039/501100011033/ "ERDF A way of making Europe", EU; by the Project VUL-NEUREA Grant PID2021-122290OB-C21 funded by MCIN/ AEI / 10.13039/501100011033 / "ERDF A way of making Europe", EU; and partially funded by the Community of Madrid (S2018/EMT-4362) SEGVAUTO-4.0-CM.

**Informed Consent Statement:** Informed consent was obtained from all subjects involved in the study.

**Data Availability Statement:** Not applicable.

**Acknowledgments:** This study benefited from the research activities developed by INSIA-UPM and CEDINT-UPM within the OPREVU project, VULNEUREA project and SEGVAUTO-4.0-CM scientific programme. The authors would like to thank the experts of the Spanish Traffic Police and the Spanish Traffic Directorate (DGT) for their contribution.

**Conflicts of Interest:** Ángel Losada reports financial support and administrative support were provided by Universidad Politécnica de Madrid. Francisco Javier Páez, Francisco Luque, and Luca Piovano report a relationship with Universidad Politécnica de Madrid that includes board membership.

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
