# Peer review of "Application of Machine Learning Techniques for Predicting Potential Vehicle-to-Pedestrian Collisions in Virtual Reality Scenarios"

_applsci, doi:10.3390/app122211364_

Round 1
Reviewer 1 Report
This is an interesting research work having the potential to reduce accidental damage. Hence, this work can have significant effect in saving precious lives and infrastructure. I would recommend accepting the paper with minor updates.
1) Material and Method section needs an introductory paragraph explaining the reasoning / justification of the subsequent subsections. Similarly, insert a brief paragraph right after Discussion section, before going into the subsequent subsections within Discussion.
2) Too many multi-level bullet points within line 270 and 303 made it confusing to the readers. Try using numeric and bullet points to distinctly represent the concepts.
3) Fix the formatting issues present within the paper. For example, Table 3 is overlapped with line numbers.
4) Clearly highlight the limitation of this study within Conclusion section.
Author Response
Dear,
The authors have introduced a brief description both in the Materials and Methods section and in the Discussion section (Point 1), expanding the content of these sections. We have replaced the bullet points of the test procedure with numbering, to make the sequence of the stages clearer (Point 2). Regarding the limitations, we have organized the text of the conclusions, unifying the part where we explain the possible improvements in terms of extending the sample, the use of Eye-Tracking, and the need to incorporate V2P communication technology to obtain information on one of the variables (Point 4).
In terms of formatting, the introduction of change control has changed the distribution of some elements in the document. We will wait for all the reviewers to evaluate the document and confirm which changes should be accepted, to proceed with the final formatting of the text, if required (Point 3).
I look forward to your feedback. Any additional changes or suggestions for changes, we are at your disposal.
Thank you very much for your time and attention, and sorry for the inconvenience.
Kind regards,
Authors (A.L., F.J.P., F.L, L.P.)

Reviewer 2 Report
Please, see the attached file.

Author Response
Dear,
Following your recommendations, the authors wish to state the changes we have made, in the same numerological order:
Point 1: We have added in the Literature Review some of the pedestrian recognition technologies in emergency braking systems.
Point 2: We have explained the reason for selecting collision scenarios with all vehicles appearing from the left side, based on information obtained directly from INSIA-UPM database.
Point 3: The phrase "are also recorded for the attempting vehicle" has been removed, as it belonged to a text that was later edited.
Point 4: A little more detail has been provided regarding the estimation of distances in the RV tests.
Point 5: The information on the calculation of correlations and the tests performed to determine the statistics has been expanded. A correction has been made on one of the statistics used. The results are the same as in the document submitted for the first time. The explanation of the selection of variables has been made clearer.
Point 6 And Point 7: Figures 11 and 12 have been corrected, trying to show the information required by the reviewer. The number of users completing the questionnaires is the same as at the beginning of section 3.3. : “The tests that make up the experimental VR session are held with a sample of 29 users with similar socio-demographic conditions...". In the same section, the test procedure is detailed, where the following is stated: "Each subject much complete the following procedure:..". Next, the questionnaires to be completed by each subject are listed (each subject must complete the SSQ -3 times-, the SUS, and the PQ).
Point 8: In section 4.2, following the reviewer's recommendations, a table has been incorporated with the hyperparameters of each model and the range of values of each one of them in which the optimization function was going to be implemented. At the end of this same section, an additional table summarizes the value of these hyperparameters, following the same format as the previous one. The explanation of the hyperparameters is contained in the text in more detail.
Point 9: A comparison of the results obtained in our algorithm for the performance metrics (accuracy, recall) with that of an algorithm integrated into a vehicle camera system to determine whether danger exists or not, introducing pedestrian behavior parameters, has been introduced, similar to that of our paper. More bibliography has been found, although in this case, it refers to the Machine Learning classification algorithm for accident types or injury severity, which is a more common topic in the literature for this type of methods.
Point 10: In Table 3 (now Table 9 after the relevant modifications), the precision and recall values for each ML model have also been incorporated.
Point 11: The authors corrected the explanation of the integration of MediaPipe and Open CV for the pedestrian face and eye detection system. Currently, the system is still under development and has not been completed, so we have corrected the paragraph by adjusting the verb tenses. In order to include it in a possible future article, we would like to optimize the system and improve the predictions obtained. Therefore, it is stated as future lines, as part of the technological development that the algorithm would need if it were to be implemented in new generation vehicle AEB systems.
We will wait for all the reviewers to evaluate the document and confirm which changes should be accepted, to proceed with the final formatting of the text, if required.
I look forward to your feedback. Any additional changes or suggestions for changes, we are at your disposal.
Thank you very much for your time and attention, and sorry for the inconvenience.
Kind regards,
Authors (A.L., F.J.P., F.L, L.P.)

Round 2
Reviewer 2 Report
I thank the authors for replaying point by point to the comments. The new information has further improved the readability and overall quality of the paper, which, in my opinion, can be accepted in its present form.
One small note: in Table 8, the range of the K parameter ([1,31]) was given for the KNN model instead of its optimal value (which should be 8).